# Sustainability of evidence-based policy engagement model: A case study of Advance Family Planning initiative in India

**Shumayla Shumayla** [ORCID]*, **Kamlesh Lalchandani, Deepali Verma, Gulnoza Usmanova, Shadab Hussain, Somesh Kumar**

Jhpiego-An Affiliate of Johns Hopkins University, New Delhi, India

* shumayla@jhpiego.org

## Abstract

### Background

Advance Family Planning (AFP) is a global engagement initiative to expand access to quality contraceptive information, services and supplies by fostering policy pledge and investment. Since it was launched in 2009 the initiate was implemented in ten countries. In India, this program was implemented in 42 districts across six states during 2012-2019. This paper describes the sustainability of its Engagement model and outlines programmatic strategies that facilitated the same.

### Materials and methods

An explorative qualitative study was conducted to capture information from the representatives of district working groups and implementation partners from 26 districts across six states. The analysis was informed by the Beery's framework for sustainability and was performed thematically using a mix of inductive and deductive approach.

### Results

The integration of district working groups with existing health system partner forums or inter-departmental coordination platforms in selected districts, and adoption of SMART advocacy approach to various other programs and beyond the initiative geographies demonstrate the sustainability of the engagement model. The factors considered essential to its sustainability include strengthening the operational capacity of district working groups through regular monitoring, creating champions by training on SMART advocacy approach, fostering multi- and inter-sectoral partnerships and networks through collaborative platforms, and promoting accountability and ownership through open discussions.

### Conclusion

The evidence emphasizes the continuity of the program as a system-led initiative for its sustainability. Health promoters and public health practitioners could proactively integrate sustainability components within their program design and achieve this through strategic planning throughout program implementation and the program lifecycle.

**Data availability statement:** This article reports findings from interviews conducted with multiple stakeholders, who consented to share their experience under condition of anonymity and confidentiality. Therefore, due to ethical restrictions, we are unable to share the complete data set. To enable replication of the findings reported in this article while adhering to our ethical obligations, all pertinent data are included in the paper as excerpts. A minimal anonymized data set can be provided upon reasonable request, with requests directed to the corresponding author or Dr. Ashish Srivastava, Country Lead-MERL and can be reached out at Ashish.Srivastava1@jhpiego.org.

**Funding:** KL has received the funding. The grant number for this award was 132328. The work described in this case study was supported by Johns Hopkins University. The website for the funder is https://jhu.edu/. The funder does not have any role in the study design, data collection and analysis, decision to publish, or preparation of the manuscript.

**Competing interests:** The authors have declared that no competing interests exist.

## Introduction

Family planning (FP) significantly contributes to multiple Sustainable Development Goals (SDGs), making it a vital aspect of healthcare [1]. It promotes maternal and child health outcomes by averting unintended pregnancies and fostering birth spacing; reduces poverty and hunger; encourages gender equity and women empowerment, combats HIV/AIDS, and ensures environmental sustainability [2,3]. Adopting and implementing exemplary initiatives over the years by Government of India has led to the significant evolution of FP program which has broaden the scope of services and shift its focus from population stabilization to a rights-based approach [4–6]. At the London Summit on Family Planning, India pledged to improve the reach of the FP services and was further revitalized to make the health system more responsive to local needs [7]. To advance its commitment, a comprehensive action plan was developed by leveraging a decentralized planning approach ingrained in the Constitution.

In a decentralized health system, national policies are formulated by the centre and districts bear the responsibility for planning and executing them, often posing implementation challenges. Globally, advocacy has been proposed as an imperative action to address enduring impediments to policy implementation and effectively achieve national commitment at scale [8,9]. It brings evidence into the policy-making process as well as engages decision-makers and stakeholders, eventually leading to a much-needed focus and resources to build and sustain equitable health systems [9,10].

Realizing the role of strategic efforts in bringing change, Bill & Melinda Gates Institute for Population and Reproductive Health in collaboration with the Johns Hopkins Bloomberg School of Public Health launched the Advance Family Planning (AFP) initiative in 2009. It was a global initiative that engaged multiple partners from various countries, including Burkina Faso, the Democratic Republic of Congo, India, Indonesia, Kenya, Nigeria, Senegal, Tanzania, and Uganda. It aimed at expanding access to quality FP services by fostering policy pledges and financial investment [11]. The literature suggests that sustained advocacy efforts are critical for advancing a favourable environment for implementing FP policy and ensuring long-term impact [12,13].

Measuring the sustainability of public health program interventions have received mounting interest globally [14]. Chambers et al. define it as the continuous delivery of intervention over time and institutionalized within settings, with the adequate capacity built to facilitate their continuation [15]. The sustainability of public programs is vital as sustained programs are capable of upholding their effects over a prolonged period and promote benefits even after the program ends [16–18]. Without sustainability, there would be a loss of investment for both organizations and the people involved [12,19–21].

In this paper, we intend to describe the sustainability status of AFP engagement model and strategies that contributed to it, drawing evidence from its implementation in India. The findings will hold particular relevance for decentralized health systems, where this engagement model could strengthen the implementation of other health programs as well.

### Advance family planning in India

In India, AFP was implemented from 2012-2019 in selected 42 districts across six states (Assam, Bihar, Rajasthan, Jharkhand, Uttar Pradesh and Maharashtra) with technical assistance from four in-country partners - the Foundation for Reproductive Health Services India (FRHS), Jhpiego India, Pathfinder India, and Population Foundation of India (PFI) (Table 1). These states and districts were given precedence because of the existing relationships of the implementing partners with the health leaders and their commitment to enhancing family planning indicators in their region.

**Table 1. Implementation partners for Advance Family Planning in each intervention state.**

| Implementation partners | Intervention states | Number of Intervention districts |
|---|---|---|
| **Jhpiego** | Assam | 4 |
| | Jharkhand | 7 |
| | Maharashtra | 2 |
| | Uttar Pradesh | 4 |
| **Population Foundation of India (PFI)** | Uttar Pradesh | 12 |
| | Bihar | 6 |
| **Foundation for Reproductive Health Services (FHRS)** | Rajasthan | 1 |
| **Pathfinder** | Rajasthan | 6 |
| **Total** | | **42** |

The initiative-supported engagement efforts could be regarded as having accelerated the implementation of FP policies in the intervention districts across six states. The incremental changes achieved - referred to as 'quick wins' - included policy wins that expanded the reach of FP methods and delivery of quality services; financial wins wherein funds were mobilized for strengthening health facility infrastructure and train service providers; and visibility wins which enhanced the awareness and prominence of family planning. During seven years of the program, a total of 324 quick wins were achieved at the national and sub-national levels [22].

## Materials and methods

### Study design

An exploratory qualitative study was conducted during February 2023-April 2023, after two years the initiative concluded. The assessment was guided by Beery's Framework that allows to evaluate the sustainability of an initiative beyond its funding period and the factors contributing to it [23]. The key constructs of the framework include initiative and its components, transition activities for conclusion of the initiative, intermediate outcome (sustainability), long-term outcome of sustainability efforts, and influencing factors. Since the study focussed on assessing the sustainability of program components post-initiative phase and not the long-term outcomes, i.e., health outcome or services, we limit description of the findings to the construct of intermediate outcome.

The qualitative study included semi-structured Key-Informant Interviews (KIIs) and Focus Group Discussions (FGDs) with the representatives of district working groups and development partners, who were actively involved in strategic planning or implementation of the mentioned initiatives at the state or district level.

### Study setting

The interviews were conducted in a representative sample of districts, i.e., 26 out of the total 42 districts across the intervention states. These districts were selected systematically using a composite index, computed using National Family Health survey 5, 2019-21 [24] variables related to family planning - total modern method uses, total unmet needs, and proportion of current users informed about the side effects of FP methods. Within each state, districts scoring higher on the index were categorized as high-performing districts, while those scoring lower were designated as low-performing districts. Subsequently, 26 districts were selected purposively for the study, ensuring equal representation of geography, high-performing and low-performing districts within each state, and implementation partners.

## Sampling, recruitment and participants

The respondents were selected purposively to ensure diverse perspectives were captured on program components and their contribution to sustainability. They were approached for the study either by telephone or with the assistance of point-of-contact individuals within specific departments or organizations. Since the conclusion of the program, many district working group members who were engaged in the implementation of the program were either reassigned to different positions in different states or have retired from their roles. This posed a difficulty in engaging the number of respondents required for FGD. However, considering the specificity of the topic and homogeneity of their role, we decided to proceed with the discussion even if 3-4 members were available, aiming to gather focused and insightful feedback. In instances where even minimum number of members were unavailable within a district, KIIS were conducted as an alternative. In total, we conducted 44 KIIs and 10 FGDs across the selected districts, as mentioned in Table 2.

## Data collection

The interview guides were developed around the thematic area of the program to be explored and were informed by the research question under consideration. Broadly, it explored the experience of the respondents with the implementation of AFP program, functioning of district working group, applicability of SMART advocacy approach, program attributes that facilitated sustainability of the program engagement model. The interview guides were pilot tested prior to data collection.

Both, KIIs and FGDs were conducted in person, at a place and time convenient to the respondent(s). KIIs typically spanned around 30-45 mins while FGD lasted 60-90 minutes. These were moderated by two public health researchers in the language preferred by the respondent(s) - English, Hindi or the local language of their respective state; and were digitally captured using audio recorder. Regular debriefing sessions were conducted with the moderators to amend interview guides based on the emergence of new information or refine lines of inquiry. The data collection continued until theoretical saturation was attained.

## Data processing

The audio recordings were transcribed verbatim and translated into English. To ensure the accuracy and completeness of the transcription, two researchers who were part of the program, reviewed the transcripts while referencing the corresponding audio files. Any discrepancies identified were resolved through discussion with the transcribers. Moreover, each transcript was anonymized to protect participant's privacy and data confidentiality.

## Data analysis

The transcripts were thematically analysed inductively and deductively using Braun and Clark's approach [25]. Moreover, we adopted interpretative phenomenological epistemology

**Table 2. Number of KIIs and FGDs conducted across the selected districts.**

| Stakeholder | Data collection level | Data collection method | Count KIIs/FGDs | Data collection approach |
|---|---|---|---|---|
| DWG members | District level | KII | 36 | **Priority approach:** If 3-4 DWG members were available within the district, FGD was conducted |
| | | FGD | 10 | **Alternative approach**: If the minimum required participants were unavailable, KIIs were conducted |
| Development partners | State and National level | KII | 8 | NA |

paradigm, emphasizing the subjective experience of the participants and the interpretation they attribute to the program components.

We ensured analytical rigor by employing a comprehensive audit trail and utilizing a multi-coder approach. This allowed iterative and consensus-driven analysis. Initially, three researchers and a program expert thoroughly read three randomly selected transcripts to gain familiarity with the content of the transcripts. An initial coding framework was developed following the inductive and deductive approaches as mentioned. Subsequently, researchers independently coded another five transcripts following the coding framework developed. A series of consensus meetings were conducted in the presence of the program expert, to discuss the discrepancies and reach an agreement in the coding process. The pre-existing and new themes were compared to refine the framework, which encompassed the addition of new themes, deletion of redundant themes, and integration of overlapping themes. This iterative approach allowed team to exclude the constructs and/or subconstructs that were not relevant for the study or didn't emerge during the process. After achieving a satisfactory level of agreement, the transcripts were coded individually. The textual data was organized under relevant themes and presented as 'participant quotes' in Italics. The data was analysed with the help of Dedoose software. The Standards for Reporting Qualitative Research guidelines were followed to ensure rigorous reporting of the study [26].

### Reflexivity statement

The coding team has research experience in the field of family planning and reproductive health, implementation science, public health and social science. Moreover, they are skilled in primary research including quantitative and qualitative methods. The team reflected on personal biases or judgement throughout the process of analysing. This allowed them to reflect on the emerging themes through different perspectives and to construct meaning in the synthesis. To further increase reflexivity, 'negative' cases – those that conflicted with or offered alternative explanations to emerging themes - were carefully considered.

### Ethical considerations

The study was performed in accordance with the ethical standards in 1964 Declaration of Helsinki and approved by the institutional review board and ethics committee of John Hopkins School of Public Health (IRB No. 23405) and Sigma (IRB No. 10099), India. Written informed consent was obtained from each respondent for their participation and audio recording. The electronic versions of project documents and data were stored on a system in password-protected folders, while physical copies were maintained in a locked cabinet to ensure data confidentiality.

## Results

A total of 44 KIIs and 10 FGDs were conducted across the selected districts. The FGDs were conducted only with the DWG members while KIIs were conducted with DWG members and development partners. Using the key insights from FGDs and KIIs, the findings were synchronized using Beery framework around components of the initiative, activities during the initiative, transition activities during conclusion of the initiative and intermediate and sustainability outcomes.

### Assessing program sustainability using adapted version of Beery framework

### Components of AFP initiative

AFP initiative employed a comprehensive engagement strategy, including formulation of district working groups and using the SMART (Specific, Measurable, Attainable, Relevant, and Time-bound) approach.

a. The District Working Group

The district working group was a collaborative platform within each intervention district, where diverse stakeholders involved in the provision of FP services were brought together to effectively identify and address local challenges to FP service provision. Stakeholders identified included health and non-health officials from the existing system; representatives of medical associations, civil societies and non-governmental organizations; and community members. Implementation partners sensitized the stakeholders on the importance of taking the family planning agenda forward, provided training to them on the SMART advocacy approach, and briefed them on their respective roles.

b. SMART Advocacy approach

The initiative adopted SMART framework to effectively develop, implement, and evaluate a focused strategy to advance the family planning goals. It outlines nine steps divided into three phases: Build consensus, focus efforts, and achieve change. These progress from recognizing the opportunities to support family planning agenda to defining an objective, executing a strategy aimed at a specific policy or funding decision, and reflecting on the experience to address subsequent challenges in pursuit of the goal [27]. Moreover, it allows champions to break down their objectives into manageable milestones, thereby facilitating a rational approach towards achieving their overarching goal. The focus remains on achieving policy or funding decisions that have an impact in the near-term, and subsequently establish the groundwork to advance towards a broader goal [11,27].

During program implementation, the initiative partners conducted 93 SMART facilitations to build the capacity of champions on advancing family planning leading to the creation of a pool of "Champions". These sessions reached 793 administrative and health officials and more than 130 development partners and civil society organizations [22].

## Activities during the initiative

**Landscape analysis.**  The DWG members would initially identify the region-specific issues through landscape analysis and discuss them in a meeting for framing evidence-based objectives that would support family planning goals in the region. This process would then guide in developing communication strategies and messages tailored to the local context for decision-makers and preparing action plans aligned with local priorities.

*"We had first meeting. In that meeting the data was presented on family planning services and logistics. He very clearly articulated which are focused blocks, what are the focused areas, where we need to improve the services. For example, in one of the districts there were three blocks which had lower FP utilization. In another district there were two blocks which were facing logistics issues. So, in that way the initial discussion was around analysing the problem. After that meeting, we recognised what are the priority areas. Keeping those priority areas in mind, the objectives were decided. Accordingly sub-activities were planned."* **DWG member, Intervention District**

**Policy engagement.**  Following the outlined engagement strategies and messages, they would approach the potential decision-makers and provide them with the gathered evidence. The evidence-informed strategy to enhance the momentum for equitable availability of high-quality FP services and effective implementation of existing policies at national and sub-national levels.
**Monitoring.**  The DWG members would meet monthly to monitor the progress of FP objectives outlined in the previous meetings. Moreover, any operational or resource gaps were identified, and corrective actions were planned.

## Transition activities during conclusion of the initiative

The initiative partners proactively took multiple measures at the time of exit to foster its self-sustenance. For instance, one of the DWG members was designated as the convenor to ensure efficient operation of the group after integration into pre-existing health forum. This was to facilitate effective communication of the program approach and goal to new FP champions who earlier were at different position or role. Depending on the region, it was District Program Manager (DPM), Additional Chief Medical Officer (CMO) or Medical Officer (MO). Moreover, they secured separate budgets for the DWG meetings in the PIPs (Program implementation plan) under the NHM (National Health Mission) for most of the districts through initiative engagement efforts.

Moreover, a comprehensive graduation training session focused on the SMART approach was organized as an integral plan of program exit phase. This was to ensure all stakeholders are well-informed and aligned with the approach so they could effectively extend their support for family planning.

## Intermediate outcome

**Integration of District Working Groups (DWGs) with health system forums.** The stakeholder responses indicated that DWGs in all selected districts were merged with existing health system partner forums or inter-departmental coordination platforms, except a few which remain untraceable (Table 3). More than half of them (58%) have been functioning in accordance with the terms outlined by the initiative. During meetings, the FP agendas are deliberated along with other health programs such as immunization, maternal and child health, and others. The action plans for FP services are designed and monitored as before. Moreover, FP is also discussed at other platforms such as the District Indemnity-committee (DISC) and the District Quality Assurance Committee (DQAC).

The ongoing meetings and dialogues on a broader scale, without any external support, imply the concerted commitment of the family planning champions towards sustainable efforts. Moreover, the integration ensures that FP remains a key consideration in broader health discussions and decision-making processes.

*"It is held every month. Although our health partner has changed, we are still doing it monthly. We conduct meetings of Zila Swasth Samity (District Health Society, DHS), Block Swasth Samity (Block Health Society) and at lower level, we conduct sector meetings... meetings with ASHA workers. The decisions or action plans that are taken at DPIB (District program implementation body) district level meetings are told at block level of sector meeting or to the ASHAs."* **Health Officials, Intervention District.**

Table 3. Status of DWGs in 26 selected districts across six states.

| States | Merged with other health forum | | Could not be tracked | Total |
|---|---|---|---|---|
| | Working as per the original ToR | Not working as per the original ToR | | |
| Assam | 3 | 1 | – | 4 |
| Bihar | 1 | 2 | 1 | 4 |
| Jharkhand | 3 | 1 | – | 4 |
| Maharashtra | 1 | 1 | – | 2 |
| Rajasthan | 4 | – | – | 4 |
| Uttar Pradesh | 3 | 2 | 3 | 8 |
| Total | 15 | 7 | 4 | 26 |

The divergence in the functioning of certain district working groups were attributed to factors, including the shift in focus towards curbing COVID-19 during the pandemic, priority of the officials towards other programs, and transfer of officials to another department or another role.

**Institutionalization of SMART advocacy approach.** While emphasizing on the importance of the SMART advocacy approach in promoting sustainable and long-term efforts, respondents affirmed their commitment to employing their engagement knowledge and skills consistently gained during the initiative, even after integrating into a larger health forum.

"*We use the approach even today. The AFP program is over, but we still use this approach at the organizational level. We keep conducting the capacity building of staff, the entire program staff, on the use of AFP smart tools. It is not necessarily FP program team. We have our communication team, adolescent health team, community action for health team. Basically, we also have master trainers within the organization now and we keep on doing this. I am glad to share this that we are planning to start another round of AFP smart training soon for our current program staff as many new people have joined. The reason for me to share this is that, as an organization we have completely internalized how we look at this approach. So, yes, it is not just about the grant. Initially it was about grant. But when we talk about this approach, we do not look at it just as an approach for implementing the grant but rather much beyond that.*" **Program lead, Implementation Partner, National Expert**

The responses also revealed adoption of SMART advocacy approach to various other programs and beyond the initiative geographies. Their success stories demonstrate usability and effectiveness of the approach across the organizations and domains, including completion of large-scale COVID-vaccination, improvement in maternal health indicators, expansion of Comprehensive Sexuality Education (CSE) program, successful implementation of non-communicable disease program, and others. SMART advocacy has been viewed by the respondents as a fundamental approach for enhancing any health program. It was considered to serve as a guiding framework, enabling them to develop their own engagement action points and implementation strategy to promote the family planning goals. In fact, the initiative partners have ingrained this approach into their organization across various programs. They prioritize capacity building by conducting training sessions on SMART approach and creating master trainers within the organization.

## Program strategies facilitating the sustainability of the engagement model

Thematic analysis identified four programmatic attributes that led to sustainability of program engagement model: Strengthen operational capacity, create champions, foster partnership and network, and accountability and ownership. The conceptual model for sustainable engagement approach that emerged from this study is presented in Fig 1.

**Strengthen operational capacity.** The strategic and engagement efforts of the implementing partners were specific to the FP issues within their geography. However, the shared objective was to establish effective DWGs at the state or district level. The implementation partners served as a catalyst to enhance the operational capacity of DWGs, enabling them to lead promotional efforts on priority issues without any external support in the long term. During program period, they ensured meetings were conducted regularly with a complete quorum and in alignment with the SMART approach. Moreover, they tracked the progress against proposed activities for each objective and revised the course of activities when necessary. These enhanced the technical capacity of the DWGs in terms of planning,

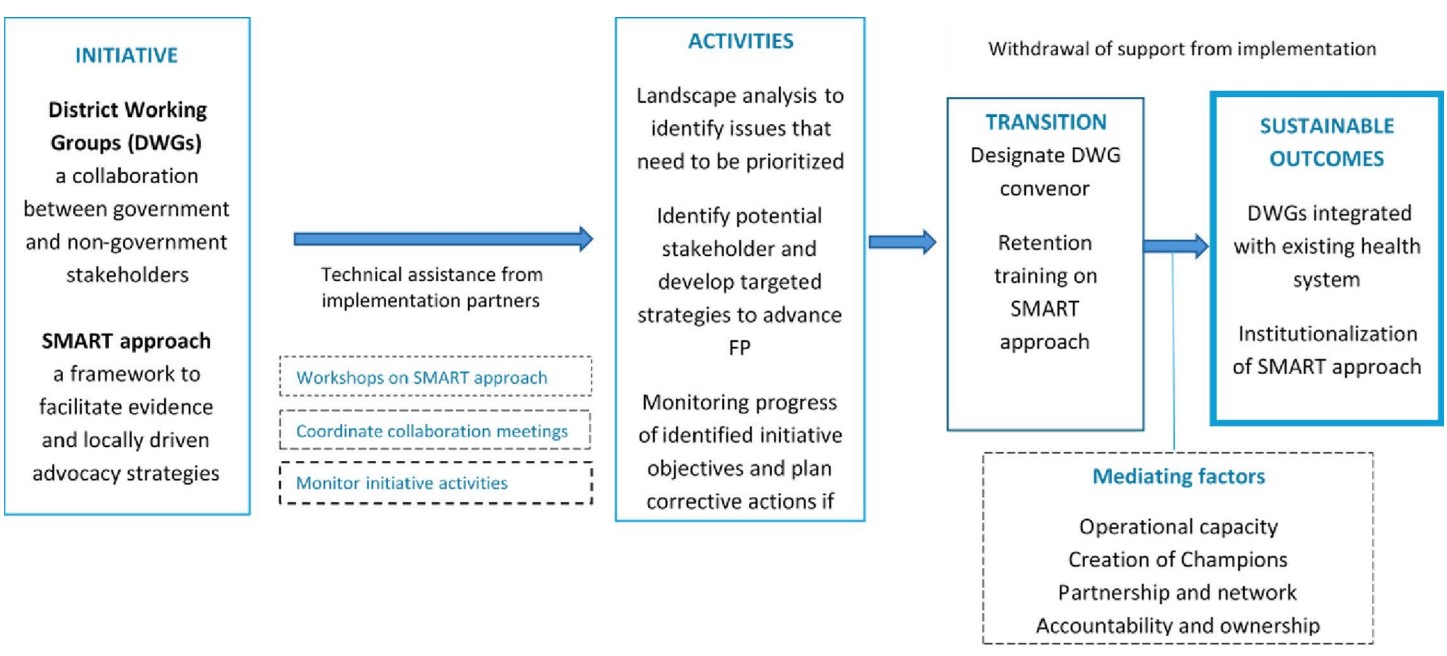

**Fig 1. Conceptual model demonstrating sustainability of AFP Engagement model.**

management, and coordination as well as fostered evidence- and locally-driven decision making – all ensuring sustainability of the initiative beyond external support.

**Create champions.** The engagement initiative created a pool of champions through regular training of the FP champions on SMART advocacy approach. These not only included the DWG members but also representatives from supporting non-government organizations or civil societies so that their engagement messages were strong. The respondents acknowledged that the training empowered them to foresee what information and insight will persuade the decision-maker to take an action towards the program goal. They were able to anticipate the opportunity to advance FP and respond quickly and decisively while leveraging partner resources. Identification of decision-makers with authority over budget and policy decisions as well as knowledge of local context facilitated development of targeted-engagement strategy. Moreover, by immersing themselves to grassroot level, the champions realized that they were able to gain a deeper understanding of the challenges and complexities surrounding FP issues. With a clear perspective, the chances of achieving goals improved.

> *"Challenges were in terms of, you know, priorities changing, but what helped was creating these champions, so when you have a district working group, they become the champions. Another example that I would give you is that there is a frequent transfer of officials, which is a huge problem in India and in most of the states. But you know, what we are doing is we are not just working with the decision maker or the leader, but we are working with a cohort of people or working group, which is constituted and formalized. So once a decision maker changes, as an organization, we never had to go, it was a working group who brought to speed. These were the champions within the systems, who were able to drive the agenda of family planning and prioritize it in the agendas of the new leadership."* **Program Implementation Partner Representative, National Expert**

**Foster partnership and network.** During discussion on the functioning of the DWGs following discontinuation of support from implementation partners, respondents affirmed that they have been employing their pre-established inter-sectoral and interpersonal relation in their ongoing efforts to promote the FP agenda. This indicates sustainability of the extensive network of collaboration and coordination established among officials and stakeholders through the initiative.

*"But once we became a part of the* initiative*, we became a part of the larger network where each partner could have supported us, and we could have supported them as well. I remember JHPIEGO, Pathfinder, and Population Foundation were our partners as well. So, this gave us the opportunity to discuss with the initiative members and other stakeholders what we want to advocate to the Ministry of Health at the National level and the provincial level and have ideas, constructive feedback as to how we can approach it better."* **Program Implementation Partner Representative, National Expert**

**Accountability and ownership.** The engagement decisions were data-driven and tailored to local context. In contrast to large-scale surveys, the landscape assessment allowed for the identification of the FP needs and challenges specific to the region and during that timeframe. The decisions were made at every stage using evidence to inform the support required to advance common FP goals and implementation strategy. It was a two-way communication – ground to the district level and vice-versa. The block level officers conveyed the grassroot issues to the district level officials. These issues were then discussed in the initiative review meeting, from the perspective of various sectors involved. With gathered feedback, engagement and implementation strategy with clearly defined role and responsibilities of the stakeholder involved was planned. The action plan was then communicated back to block level official and further down to the service providers at ground level. The responses suggest this iterative communication and feedback loop helped FP champions in addressing issues effectively and coordinating actions across different levels of the program implementation.

*"When this group was formed, at that time we all realized that if the collector asks me any question what my answer would be; because he could ask individually what you did or what kind of problem you are facing during the implementation. And I think, that urged us to work hard throughout the month."* **Health Officials, Intervention Districts.**

## Discussion

This study demonstrates sustainability of the Advance family planning initiative, achieved through integration of DWGs into health system-led forums and institutionalization of the SMART approach. Assessing the sustainability of health intervention is pivotal for public health evidence users like practitioners, policymakers and funders. Stakeholders want to understand if the benefits of the interventions will be sustained beyond the funded period and intervention's lifespan [28]. Through DWGs integration, the program has ensured its continuity as system-led and owned initiative. Ownership fosters collaborative decision-making, ensure stakeholders are held accountable, and preventing fragmented efforts while optimizing resource utilization [29,30]. Institutionalization of SMART approach has led to diffusion of approach to other health and development program and beyond the program geographies. This serves as a fundamental mechanism for fostering accountable and transparent institutions, that are also resilient and sustainable." [31,32]. Furthermore, there are evidences of effective utilization of this approach in healthcare from other countries, particularly Asian

and African [33,34]. These highlight the adaptability of the SMART approach to any setting or context, which has been considered to promote sustainability [35].

The findings emphasize the incorporation of parameters promoting sustainability of the engagement initiative, which was embedded within the program design and its strategic implementation aligning with the project level process factors required for sustainability [18,35]. This included strengthening operational capacity of district working group to strategically implement and monitor the progress of family planning objectives, creating champions to develop and initiate evidence- and locally-driven engagement, multi-sectoral collaboration for efficient utilization of resources, and fostered accountability and ownership. These have been identified as sustainability constructs within the existing healthcare literature [36–38]. Capacity building empowers the local governance with knowledge and skills to identify and manage the opportunities and responsibilities that come with decentralization, independent of external support [39]. Within this context, the presence of champions that advance common goals within the systems has been an instrumental factor in driving the priorities and ensuring that FP remains on the agenda despite changes in leadership. This aligns with 'diffusion of innovations' which theorizes how engaging influential individuals could facilitate change in a policy or program and institutionalize it at multiple levels. This strategy has been adopted in multiple global health programs but notable for reproductive health, where champions promote to reshape and mobilize support for family planning programs [40–42]. These champions promote evidence- and locally-driven practices, making them more likely to sustain [27,43].

Multisectoral collaboration has been considered crucial for reduced health inequalities stemming from limited access and availability of quality services or socio-cultural and geographical barriers. However, uncoordinated efforts due to differences in institutional priority, undefined roles and responsibilities, lack of shared vision and leadership determination, limited accountability, and poor strategic capacity remain a challenge. These have often led to redundant efforts and unconstructive collaboration [34,44–46]. AFP initiative addressed these challenges by creating a conducive environment for partnership and networking, i.e., by sensitizing the stakeholders from different sectors on FP program goals, providing training to build their capacity in promotion of common goals and community engagement, and involving them in decision-making. These strategies have also been considered effective by other scholars in addressing challenges faced in the pursuit of sustainability [47–49].

Moreover, SMART approach has empowered the champions in identifying the decision-makers with authority over financial or policy change and developing focused communication strategy. This has eventually strengthened their potential for creating future partnerships. Additionally, the collaboration nurtured by the SMART advocacy has not only fostered valuable partnerships and networks for the long term but also facilitated stakeholder to progress towards common goals [27]. This is becoming a preferred "mode of choice" in dealing with complex problems and aiming for sustainable effectiveness [50].

Mobilizing multiple stakeholders and fostering a robust network, broaden the scope of functioning within the system and enhance the reach of the program and policy [41]. Above all, the initiative through its collaborative platform and SMART approach, allowed the stakeholders to reach consensus on the priorities while ensuring their strategic interests meet [51]. This has been considered to foster inclusivity and ownership which eventually lead to effective and sustainable outcomes [34]. This approach aligns with the India's FP2030 vision, emphasizing collaboration, convergence, and inclusivity to provide quality family planning services through improved health systems and community engagement [52].

Locally-led engagement and review meetings played a crucial role in fostering accountability and ownership among stakeholders involved in the program. Open discussions on

the gaps and challenges promoted transparency, encouraged collaborative problem-solving, and created a shared sense of responsibility towards achieving the program goal in the long run. These two parameters among many have been considered essential for the sustainable scale-up of health programs [53].

## Limitations and strengths

The evidence for sustainability of the program is derived from the qualitative data and program exit documents. While the findings may be inherently subjective, deducing them onto a theoretically informed framework strengthen the evidence and allow generalizability. However, the qualitative data was gathered two years after conclusion of the initiative, posing challenges in identifying and approaching the DWG members or development partner representative involved during the implementation period. Their transfer to a new role or region limited their availability, and the responses might be influenced by recall bias. We interviewed respondents from selected districts; therefore, the results are not representative of the entire intervention geography. However, we employed maximum variation sampling to capture the geographic variation and a wider range of perspectives from stakeholders involved in the initiative. It was ensured that the selected sample provide richly textured understanding of the program constructs under consideration.

## Conclusion

This study builds evidence of the sustainability of the initiative that advances the FP agenda, emphasizing that this was ingrained within the program design and strategic implementation. Firstly, the integration of DWGs into existing system structures has ensured FP remains a key consideration in broader health discussions and decision-making processes. Another cornerstone has been the institutionalization of the SMART approach and its transcendence to other health programs, both within and beyond the initiative geographies. The program model, involving evidence- and locally-driven engagement and a collaborative platform, in itself, has contributed to a pool of FP champions with a sense of collaboration, accountability and ownership which are crucial sustainability factors of any health program. The evidence suggests that health promoters and public health practitioners could proactively integrate sustainability by actively planning throughout program implementation and the program lifecycle.

## Acknowledgments

We express our profound gratitude to the health system authorities for their invaluable support and cooperation throughout the study period. We extend heartfelt thanks to the participants whose willingness to share their experiences and insights has enriched the depth and relevance of our study. Additionally, we acknowledge the generous contributions from our esteemed donors.

## Author contributions

**Conceptualization:** Shumayla Shumayla, Kamlesh Lalchandani, Gulnoza Usmanova, Shadab Hussain, Somesh Kumar.

**Data curation:** Shumayla Shumayla, Deepali Verma, Shadab Hussain.

**Formal analysis:** Shumayla Shumayla, Deepali Verma, Gulnoza Usmanova, Shadab Hussain.

**Funding acquisition:** Kamlesh Lalchandani, Somesh Kumar.

**Investigation:** Shumayla Shumayla, Kamlesh Lalchandani, Shadab Hussain.

**Methodology:** Shumayla Shumayla, Kamlesh Lalchandani, Deepali Verma, Gulnoza Usmanova, Shadab Hussain, Somesh Kumar.

**Project administration:** Kamlesh Lalchandani.

**Resources:** Shumayla Shumayla, Kamlesh Lalchandani.

**Supervision:** Kamlesh Lalchandani, Deepali Verma, Gulnoza Usmanova, Somesh Kumar.

**Validation:** Shumayla Shumayla, Deepali Verma, Gulnoza Usmanova.

**Visualization:** Shumayla Shumayla, Deepali Verma, Gulnoza Usmanova, Shadab Hussain.

**Writing – original draft:** Shumayla Shumayla, Deepali Verma, Gulnoza Usmanova.

**Writing – review & editing:** Kamlesh Lalchandani, Somesh Kumar.

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
