## [Decision Letter · Decision Letter 0]

18 Feb 2025

Sustainability of evidence-based policy Engagement model: A case study of Advance Family Planning initiative in India

PONE-D-24-39917

Dear Dr. shumayla,

We’re pleased to inform you that your manuscript has been judged scientifically suitable for publication and will be formally accepted for publication once it meets all outstanding technical requirements.

Kind regards,

Philipos Petros Gile, MA

Academic Editor

PLOS ONE

Reviewers' comments:

Reviewer's Responses to Questions

**Comments to the Author**

1. Is the manuscript technically sound, and do the data support the conclusions?

Reviewer #1: Yes

Reviewer #2: Yes

2. Has the statistical analysis been performed appropriately and rigorously? 

Reviewer #1: Yes

Reviewer #2: Yes

3. Have the authors made all data underlying the findings in their manuscript fully available?

Reviewer #1: Yes

Reviewer #2: Yes

4. Is the manuscript presented in an intelligible fashion and written in standard English?

Reviewer #1: Yes

Reviewer #2: Yes

5. Review Comments to the Author

Reviewer #1: You are done a good work. also arrange well ............................................................................................................................................................................

Reviewer #2: Well structured and executed project, concentrating on the much un-met needs of the country. The methodology, analysis, results and discussion are clear without any ambiguity. Justifies the ethical considerations of the research participants.

6. PLOS authors have the option to publish the peer review history of their article (what does this mean? ). If published, this will include your full peer review and any attached files.

**Do you want your identity to be public for this peer review?** For information about this choice, including consent withdrawal, please see our Privacy Policy .

Reviewer #1: No

Reviewer #2: **Yes: ** Dr S Lokesh

---

## [Editor Report · Acceptance letter]

PONE-D-24-39917

PLOS ONE

Dear Dr. shumayla,

I'm pleased to inform you that your manuscript has been deemed suitable for publication in PLOS ONE. Congratulations! Your manuscript is now being handed over to our production team.

Kind regards,

on behalf of

Dr. Philipos Petros Gile

Academic Editor

PLOS ONE